# Permanent Makeup Removal Severe Complication—Case Report and Proposal of the Protocol for Its Management

**DOI:** 10.3390/jcm13185613

**Published:** 2024-09-22

**Authors:** Weronika Pióro, Bogusław Antoszewski, Anna Kasielska-Trojan

**Affiliations:** 1Contour Clinic (Private Practice), Ul. Legnicka 55/UA 5, 54-203 Wrocław, Poland; 2Plastic, Reconstructive and Aesthetic Surgery Clinic, Institute of Surgery, Medical University of Lodz, 90-419 Lodz, Poland; boguslaw.antoszewski@umed.lodz.pl

**Keywords:** permanent eyebrow makeup, hypersensitivity reaction, pigment allergy, secondary allergy, case report

## Abstract

**Background:** The purpose of this paper is to report the case of a patient who developed a local allergic reaction to the pigment used for permanent eyebrow makeup. In addition, the authors’ protocol for the treatment of such a complication is proposed. **Methods:** A patient visited the clinic to remove permanent eyebrow makeup. In the patient’s opinion, the eyebrows were too dark and incorrectly shaped. Upon physical examination of the brows, they were found to be over-pigmented and drawn outside the area of hair growth of the natural brow. **Results:** The patient underwent 24 treatment procedures to remove the pigmentation including four sessions of laser therapy and subsequent treatments using a chemical substance (remover). No adverse effects occurred during the initial phase of makeup removal (black pigment removal), but an allergic reaction appeared with the removal of the red-orange colored pigment (after the fourth laser therapy session). The following symptoms appeared: redness, swelling, and the appearance of papules filled with both serous fluid and pigment used for the permanent makeup. After each remover treatment, the allergic reaction decreased, and local symptoms gradually subsided. Additionally, topical corticosteroid treatment was implemented peri-procedurally. The patient’s case suggests a secondary allergic reaction to the red-orange pigment used for permanent makeup. **Conclusions:** The literature on this topic is scarce; therefore, we present a regimen for the management of such complications. In addition, we draw attention to the fact that allergic reactions to pigment may not always occur immediately following skin pigmentation but may become apparent long after the procedure, such as when the allergen is “exposed” during an attempt to remove or correct the makeup.

## 1. Introduction

A tattoo is a permanent drawing within the skin and is one of the forms of body modification performed by inserting pigment into the dermis using needles or other sharp instruments. Tattooing—as defined by the latest European regulation (EU REACH)—includes all procedures for both tattooing and permanent makeup (PMU) regardless of the technique and depth of ink insertion into the skin [1]. The technique of cosmetic tattooing is called micropigmentation. This method involves introducing micro ink droplets into the upper layers of the dermis using a traditional tattooing device or microblading machine [2].

The main indication for PMU includes an alternative to conventional makeup (lips, eyebrows, eyelines) in order to save time, when its performance is difficult, e.g., because of motoric or visual disabilities, or when a person is allergic to traditional cosmetics. The second indication includes camouflage and/or reconstructive purposes (nipple–areola complex, birthmarks, alopecia/vitiligo, and scars) [3].

Complications that may occur after PMU are most often related to the insertion of a microneedle in the skin and include bleeding, itching, and edema, but there are also reports of rare complications sometimes appearing many years after the procedure. These include granulomatous pseudotumor, loss of lashes, margin irregularities, dry eye syndrome, corneal pigmentation, corneal defect/injury, conjunctivitis, granulomatous inflammatory reaction, trichiasis, extensive hyperpigmentation of the nasojugal fold and lower eyelid and even eyelid necrosis, magnetic resonance imaging complications, and dissatisfaction with the results [3,4,5,6,7,8,9,10,11]. The study performed by Tomita et al. [12] included 1352 respondents who underwent permanent makeup procedures for the eyebrow and eyeline. They reported an overall complication rate of 12.1%. However, in most cases, complications were related to the reaction of needle passage, including itching of the eyebrow (8.2%) and swelling of the eye-lined skin (13.2%). The most severe complications were local infections (0.2%). No allergies to the used inorganic pigments were detected.

When the result of PMU is not satisfactory but permanent, the two main methods used to remove PMU are laser therapy and/or a chemical remover. Their effectiveness is dependent on the pigments used (Table 1). PMU removal requires fragmentation of permanently “embedded” particles in the dermis. The gold standard for tattoo/makeup removal is laser therapy. Short pulses of high-intensity light are applied to the tattooed skin surface. The laser light penetrates the skin and is selectively absorbed by the pigment. The absorbed laser light causes heating and fragmentation of the particles. Because of the chemical composition of various tattoo pigments, the effectiveness of the fragmentation processes is often unpredictable. To remove pigment particles effectively, a given wavelength must be absorbed by the pigment. The pulse duration should be sufficiently short (measured in nanoseconds or picoseconds) and the light energy sufficiently high [13,14]. The main factor determining the treatment parameters and wavelength is the color of the tattoo. The wavelength that is effective for the removal of black, gray, and blue is 1064 nm (Nd-YAG). This wavelength also demonstrates high safety in relation to epidermal melanin, as the longer wavelength penetrates deeper, bypassing chromophores in the superficial layers of the skin [15,16]. On the other hand, the effective wavelength for the removal of red and orange pigment is 532 nm. This wavelength, however, is not recommended for patients with higher phototypes because of the risk of skin discoloration or depigmentation [16,17]. Once the appropriate wavelength is used, the laser light penetrates the skin and reaches the chromophore, i.e., the pigment, which is dispersed in different depths of the dermis [13]. The pigment particles, broken up by the light, are released from lysosomes, macrophages, fibroblasts, and mast cells into the extracellular space and then undergo phagocytosis by macrophages and are transported to the lymph nodes. However, this chain of events may differ as the change in the physicochemical properties of the pigment and their behaviors in the process of phagocytosis is not fully known. As reported by Serup et al. [18], laser treatment alone rarely causes an allergic reaction to the pigment, and usually takes the mild form of a localized, papular dermatitis. In such cases, the patient should be prepared with oral steroid medications taken 1–2 months prior to each treatment as well as antihistamine drugs taken one hour before the laser treatment. However, Szczerkowska-Dobosz [13] claimed that an allergic reaction to pigment is a contraindication for laser treatment because of the risk of photochemical reaction of pigment decomposition and systemic reaction. According to the author, premedication with steroids or antihistamines can only reduce but not prevent an allergic reaction. Another aspect making it difficult to eliminate unwanted permanent makeup is the so-called “resistant (*to the nanosecond laser*) colors”. These pigments include chromium oxides responsible for a greenish-blue color [19,20,21]. In addition, skin subjected to permanent makeup often has scars in its structure, which make it difficult to evacuate the fragmented pigment particles. In such cases, procedures using a chemical remover are resorted to. They involve using a needle or a permanent makeup device to introduce a substance into the skin that causes chemical extraction of the pigment encrusted in the dermis. The reaction involves a complex series of changes in the skin, including epithelial and dermal damage, bleeding, inflammation, and neutrophil infiltration, followed by the formation of scabs, which consequently also causes the removal of pigment along with them [22].

### Purpose

The purpose of this study was to describe the presentation of a rare complication occurring during permanent makeup removal and the introduction of the authors’ treatment regimen for this type of complication.

This study was approved by the Bioethical Committee of the Medical University of Lodz (RNN/173/22/KE).

## 2. Case Report

A 38-year-old Caucasian woman came to the aesthetic medicine Contour Clinic for the removal of unsatisfactory permanent eyebrow makeup. The makeup was unnaturally dark, and the shape of the eyebrows was unacceptable to the patient. The eyebrows were pigmented four times at intervals of several years (Figure 1A). The patient reported no medical conditions and was not taking any medications on a regular basis. Her medical history revealed that after the third pigmentation, small itchy skin lesions of a recurrent nature were noticed on the right eyelid. After dermatological consultation, topical corticosteroid ointment treatment was implemented. In addition, deterioration of the general skin condition throughout the face in the form of increased erythema, papules, and pustules, also of a recurrent nature, was noted. The patient’s physician did not associate these symptoms with the presence of permanent makeup. Based on the patient’s medical history, the dermatologist diagnosed contact eczema and an exacerbation of rosacea especially on the cheeks, and recommended skin tests and further observation. Skin patch tests showed no abnormalities. The patient qualified for a permanent makeup removal procedure according to the protocol described below. However, the protocol was modified when a local allergic reaction appeared in the process of permanent makeup removal, including a significant deterioration of the general skin condition, i.e., increased erythema throughout the face, accompanying papules and pustules, as well as lichenoid lesions on the right eyelid. The aforementioned skin symptoms slowly resolved with subsequent permanent makeup removal treatments with a chemical remover and Nd:yag Q-switch laser.

### Treatment Protocol

Makeup removal began with the use of a Q-switch nanosecond laser with a wavelength of 1064 nm, spot size 4 mm, 3.2 J/cm^2^, 6 Hertz (Laser Xlase Plus from Biotec Italia Poland). The eyebrows were very dark and had a bluish-black tint. After the first treatment, the black pigment was removed, and the eyebrows turned an intense red color a month after the treatment. Such a distinct red color may indicate that organic pigments or hybrid pigments, among others, were used for the permanent makeup. During the second treatment, a light wavelength of 532 nm, spot size 4 mm, 6.4 J/cm^2^, 6 Hertz, was used, and the same wavelength was also applied in the next two treatments. In total, the light length of 532 nm was used three times. During the third treatment, we used spot size 2.5 mm, 4.0 J/cm^2^, 6 Hertz, and during the fourth, we used spot size 2.5 mm, 8.1 J/cm^2^,6 Hertz. After the fourth treatment, the color of the eyebrows changed from red to orange and began to take on a yellowish hue (Figure 1B). At the fifth treatment, it was noticed that the pigment did not respond to the laser light, so it was decided to use a chemical remover (Bay-Bay Pigment Remover Long Time-Liner, including sodium hydroxide, with alkaline properties at pH 9.0–11.0 (Appendix A)).

The same procedure was repeated during the sixth treatment. During this period, the patient noticed the first very mild itching sensation in the eyebrow area. This itching was not a cause for concern, as the skin of the eyebrows was unchanged. A clinically visible local allergic reaction occurred after the seventh treatment. At that time, swelling and redness of the skin with foci of inflammatory papules with serous exudate were noted. At this stage, topical corticosteroid treatment was implemented, and the interval between pigment removal procedures with remover only was extended. When the chemical agent was applied, large amounts of pigment were released from the eyebrows (Figure 2). The inflammation decreased immediately after the procedure was performed. The eyebrows brightened, and the skin texture improved. On the other hand, about 2 weeks after the procedure, the color of the eyebrows darkened to a red-orange hue, and the skin became swollen and covered with inflammatory papules. The above-mentioned situation was repeated after several more treatments, the difference being that with each treatment, the color of the eyebrows became noticeably lighter, while the itching in the eyebrow area subsided after the treatment but worsened again after a few days. Then, a nanosecond Q-switch laser (Laser Xlase Plus from Biotec Italia Poland) with a wavelength of 532 nm, spot size 4 mm, 8.1 J/cm^2^,6 Hertz, was used again to enable further chemical removal of the pigment. The reaction during the procedure was as expected, and the pigment responded to the laser light well. One day after the procedure, the patient noticed increased inflammation, manifesting as cauliflower-like serous eruptions (Figure 1C), as well as very intense pruritus. At this point, treatment with antihistamines was implemented, along with topical corticosteroids. Then, after 7 days, another chemical remover treatment was implemented to take advantage of the fact that the pigment had moved superficially after the laser treatment and to remove as much of it as possible (Figure 1D). After this procedure, the itching sensation was significantly reduced. Subsequent remover treatments were performed at 3-week intervals, for a total of 24 treatments. After this series, the skin returned to its original appearance, and the itching of the skin completely disappeared (Figure 1E). Removal of the permanent makeup and treatment of its complications were also completed at this stage. The entire program of treatment is outlined in Figure 3.

Based on the Case 1 experience, a similar procedure was carried out on another patient who had a similar allergic reaction to a pigment of similar clinical presentation, additionally performed by the same linergist (Figure 4A). The treatment regimen also involved the use of a Q-switched laser in the initial phase and then the use of a chemical remover. The first symptoms of allergic reaction occurred after the second treatment with laser light (local inflammatory reaction of the skin [edema, papules, and pustules], and accompanying severe itching) (Figure 4B). In this case, pigment removal with a chemical remover (Bye-Bye Pigment Remover) was implemented after just 2 weeks. After a few days, the pruritus and local inflammation subsided, although the skin color showed that some pigment still remained in the skin. Treatments were continued until the permanent makeup had lightened significantly, with the aim to allow for the performance of new pigmentation in the future (Figure 4C).

In a follow-up at approximately 6 months post-makeup removal, each patient was satisfied with the results and had not experienced any further allergic reactions.

## 3. Discussion

In this article, we presented the case of a patient who developed a rare complication of permanent makeup removal, outlined a treatment regimen, and presented its results. The result of the intervention was the removal of most of the pigment from the eyebrows during 24 treatments performed at three-, six-, and eight-week intervals. The treatments were performed using a nanosecond laser with light wavelengths of 1064 nm and 532 nm, as well as an alkaline chemical remover. After the first five treatments, the healing period was uncomplicated. After the sixth treatment, upon which the first signs of allergy to the pigment were noted, and with each subsequent treatment, the recovery period shortened and the skin symptoms subsided, only to intensify approximately 2 weeks after the initial recovery. Because of the lack of progress in reducing pruritus, it was decided to re-implement laser therapy in order to displace the pigment subcutaneously so as to release it from the skin more quickly. Treatment with the chemical remover was then continued until the allergic reaction had completely resolved. Based on the cases, we believe that the implementation of the chemical remover treatment at an earlier stage could have an impact on reducing the dermatitis faster. Nevertheless, the type and course of an allergic reaction are indeed influenced by the amount of encrusted dye, as it is well known that the intensity of an allergic reaction depends on the amount of allergen.

Permanent makeup, similar to tattoos, can cause adverse reactions. Both pigmentation and the removal process can carry the risk of short-term and long-term complications. Removing permanent makeup may prove problematic especially if secondary allergies occur during the pigment removal process. At this point, there is very little information on how to deal with these types of reactions. Procedures for pigment removal in the case of allergenization in tattoos located on the body are, on the one hand, more difficult to manage, as they most often involve a larger area of the body, which carries a greater risk of systemic reaction [17,18,29]. On the other hand, it is easier to decide on radical surgical excision of the pigmented skin. The problem arises when the pigment allergy is present in an area such as the eyebrows. Here, excision of the skin along with the allergen and, consequently, the hair of the eyebrows would result in an unacceptable effect. An alternative solution, although undoubtedly longer to implement, is to use laser and/or chemical remover. In the Case 1 patient, the entire process of removing the allergy-inducing pigment took almost two years and, especially when using a chemical remover, was quite painful. Nevertheless, the protocol gave satisfactory results.

Contrary to red ink tattoos, there are not many reports concerning PMU allergic reactions. Four cases were described by Wenzel et al. [30], where severe and therapy-resistant skin reactions appeared in patients after lip PMU with the use of Pigment Red. The treatment included topical steroids and tacrolimus, which appeared not to be fully effective. The authors recommended the regulation and control of the substances used in PMU colorants [30]. Goldman and Wollina [26] described a massive displacement of micropigments following eyelid tattooing, which was treated with three consecutive sessions of a Q-switched 1064 nm neodymium-doped yttrium aluminum garnet (Nd: YAG) laser—the course of treatment was uneventful and effective. The fact that allergic reactions appeared in our cases could have been related to the different types of pigments and different mixtures of pigments including not only black ink. We have not found a case of allergic reaction induced by PMU removal with a laser. However, such reactions, even anaphylactic, were reported in cases of tattoo removals with red ink and with black ink only [31,32,33]. The presented case points to the possibility of similar reactions in cases of PMU in the periorbital area. Red pigment is most likely responsible for the allergic reaction that occurred, as this reaction followed the use of a light length of 532 nm, for which the chromophore is brown, red, and orange. Numerous scientific sources also report that the red color in pigmentation most often causes allergic reactions [16]. However, this is only a suspicion. As the allergen is usually unidentified, the exact mechanism of a chronic allergic reaction to permanent makeup made with a mixture that includes red ink is not fully explained [19,20,34]. The likely mechanism is that the metabolism of the ink or the reaction of the antigen with the carrier protein in the dermis over a long period of time triggers a delayed hypersensitivity reaction [2,13,18]. According to the presumed pathophysiology, chronic allergic reactions to permanent makeup can appear months or years after its application. They usually manifest as localized swellings, papules, and pustules in the permanent makeup area, along with severe itching of the skin [14,17]. Sometimes, these lesions take on the appearance of contact eczema or rosacea and occur in close proximity to the eyebrows. Treatment of this type of reaction will most often begin with topical corticosteroids, oral steroids, and also antihistamines. Sometimes, however, a confluence of needle mesotherapy with corticosteroids is performed, especially in exacerbated hypersensitivity reactions [13,29]. Treatment of the allergic reaction in our patient involved the use of a chemical remover, although the time for complete elimination was prolonged. This form of treatment was chosen because, as reported by Serup [18], the use of a Q-switch laser may carry the risk of inducing photochemical changes in the pigment and producing new toxins and allergens. As the allergic reaction involved the skin of the eyebrows, surgical excision of the affected lesions was not considered. In addition, with each treatment, the symptoms of hypersensitivity slowly disappeared and the appearance of the eyebrows improved; thus, the patient’s well-being gradually improved, leading the patient to accept the idea of long-term treatment.

## 4. Conclusions

Pigments used for permanent makeup, as any substance administered into the skin, can carry the risk of allergy. As shown in this article, this reaction can occur right after the procedure as well as several years following pigmentation. Also, hypersensitivity to pigments can be activated by laser light. In such cases, the management of further pigment removal should be reconsidered, e.g., from Q-switch laser treatment to a chemical remover. This report also emphasizes the clinical awareness of the fact that even the smallest permanent interference with the skin can be associated with the risk of severe complications, which should be managed by professionals.

## Figures and Tables

**Figure 1 jcm-13-05613-f001:**
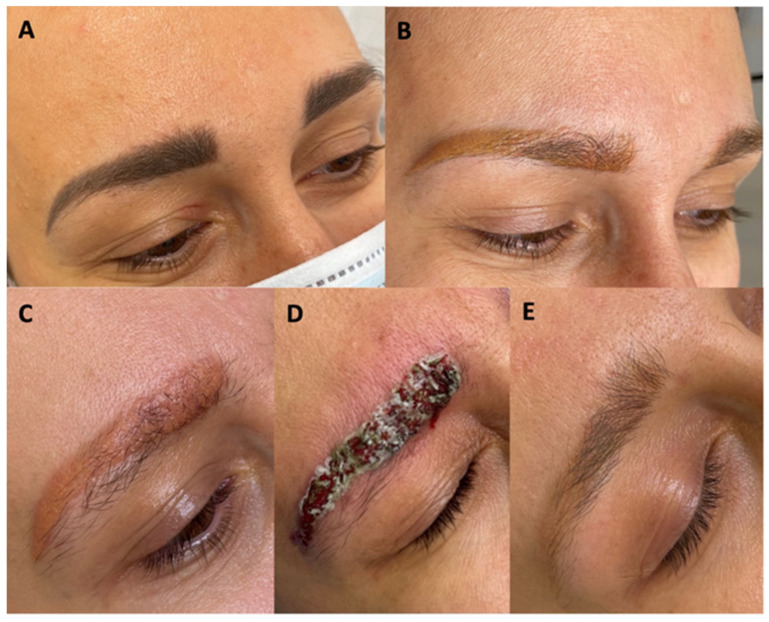
Hyperpigmented eyebrows (**A**), after laser treatments with a Q-switched 106 nm, 532 nm laser (**B**), after reapplication of a Q-switch 532 nm laser (**C**), just after the application of the chemical remover (**D**), after completing the removal process (**E**).

**Figure 2 jcm-13-05613-f002:**
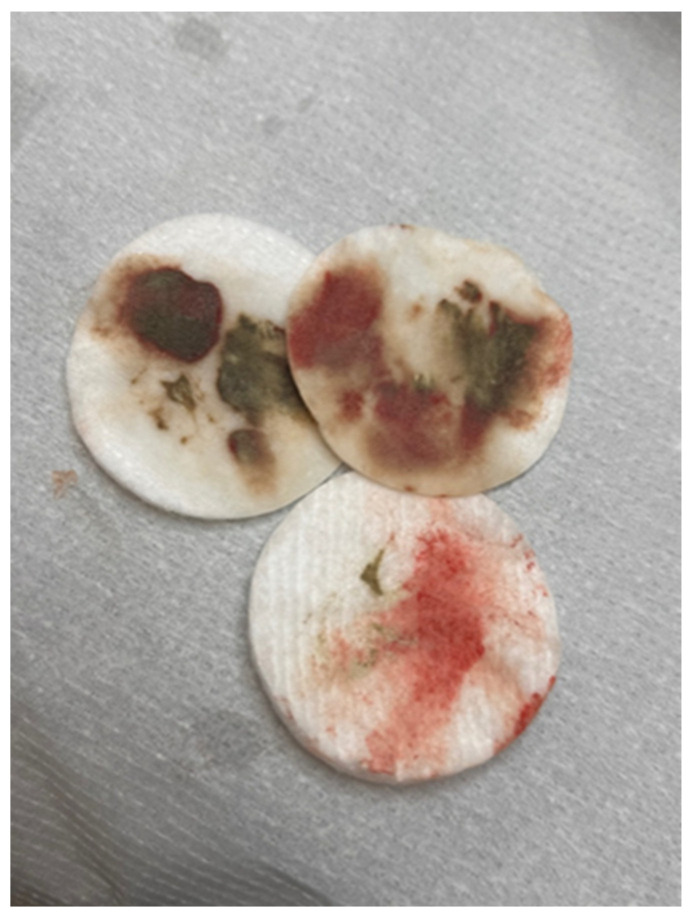
Cosmetic swabs showing the amount of pigment evacuated from the skin after one treatment with a chemical remover.

**Figure 3 jcm-13-05613-f003:**
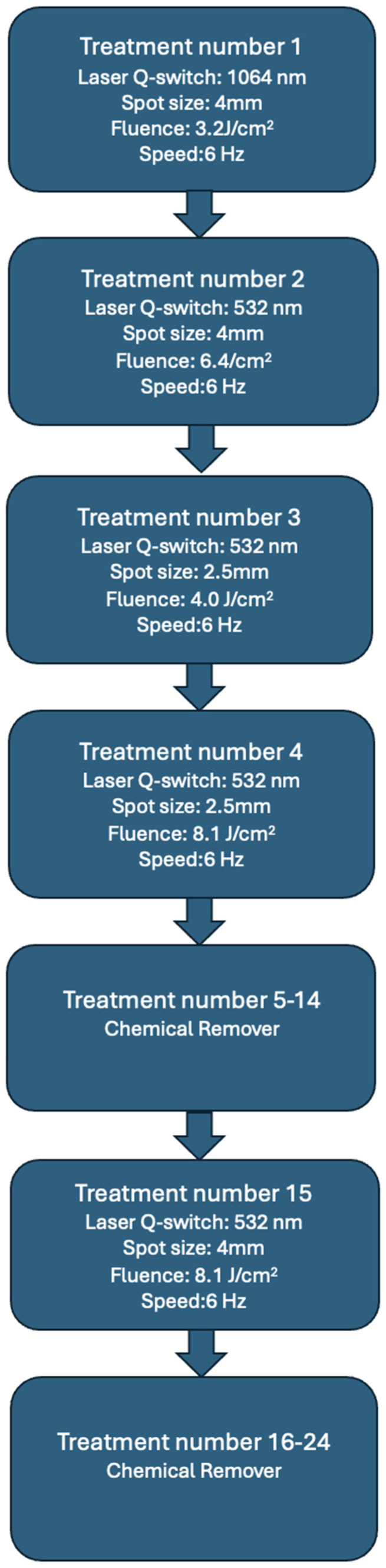
Protocol for permanent makeup removal and allergic reaction treatment.

**Figure 4 jcm-13-05613-f004:**
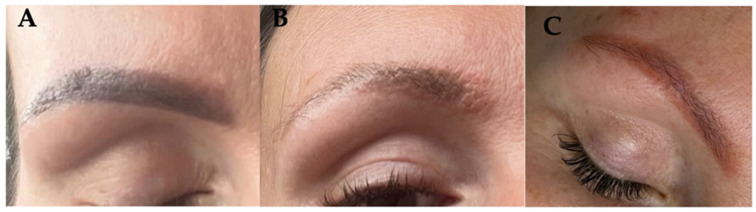
Hyperpigmented eyebrows (**A**), allergic reaction to pigment after 3 laser treatments (**B**), and after remover use (**C**).

**Table 1 jcm-13-05613-t001:** Methods used for permanent eyebrow makeup removal and/or correction *.

Method	Advantages	Disadvantages	Complications	Treatment of Complications
Laser Nd:Yag Q-switch, 1064 nm, 532 nm	Easily accessible. Low risk of complications. Effective in removing black and gray pigments.	Many treatments needed. Possible short-term and long-term complications.	Burning and itching. Oedema. Blisters. Ecchymosis and local bleeding. Discoloration. Scarring. Secondary allergy.	Topical steroids. Itch-relieving. Ointments. Hydrogel dressings. Panthenol creams to accelerate healing. Ablative lasers.
Picosecond laser	Effective in removing most dyes. Requires fewer treatments compared with the nanosec. laser.	High cost of the procedure. Severe soreness.	As above	As above
Ruby laser, 694 nm	Removes purple, green, and blue colors.	Many treatments needed. High affinity for hemoglobin.	As above	As above
Alexandrite laser, 755 nm	Removes blue and green pigments.	Many treatments needed. Not effective in removing brown, red, or orange colors.	As above	As above
Chemical Remover	Easily accessible. Cost-effective. Effective in removing all colors.	Many treatments needed. Long-term treatment. Uncomfortable for the patient/pain after application. Longer recovery time compared with laser treatment.	As above	As above

* Literature review limited to permanent makeup removal methods excluding the literature concerning tattoo removal [12,23,24,25,26,27,28].

## Data Availability

The original contributions presented in the study are included in the article (and Appendix A), further inquiries can be directed to the corresponding authors.

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
