# Peer review of "Permanent Makeup Removal Severe Complication—Case Report and Proposal of the Protocol for Its Management"

_jcm, 2024, doi:10.3390/jcm13185613_

Round 1

Reviewer 1 Report

Comments and Suggestions for Authors

The manuscript entitled "Permanent Makeup Removal Severe Complication – Case Report and Proposition of the Protocol for Its Management" deals with the report of a patient who developed a local hypersensitivity reaction to the pigment used in the application of permanent eyebrow makeup.

The paper was written with low scientific soundness. There are many mistakes:

- Lack of proper affiliations
- The laser parameters were not described appropriately, making it impossible to repeat the treatment
- There are mistakes in the description of units, such as joules
- The case description does not follow appropriate guidelines such as CARE
- Images/photos of each follow-up should be taken from the same angle

Author Response

Dear Editor and Reviewers,

Thank you for your interest in our manuscript entitled "Permanent makeup removal severe complication – case report and proposition of the protocol for its management”. We would like to thank Reviewers for their valuable comments and reviews, which helped to improve the manuscript. In this revision we addressed all your comments (in the text changes are marked in red). Along Reviewers’ comments we performed extensive editing of the paper – it was proofread by the native and references were updated and organized. All insertions are marked in red /stylist and grammar changes were not marked as they were too extensive and complex/. We are sorry for not checking this before initial submission. We hope that now, the paper reads well and will meet the requirements of the Reviewers.

Sincerely,

The Authors

Reviewers’ comments:

Reviewer 1
The paper was written with low scientific soundness. There are many mistakes:

Thank you for pointing on problematic aspects – we extensively edited the paper and included the following corrections

- Lack of proper affiliations

Corrected. To explain: the first author (W.P.) is affiliated as PhD candidate in Plastic, Reconstructive and Aesthetic Surgery Clinic, Medial University and in her own private practice /where she treated the reported patients/
- The laser parameters were not described appropriately, making it impossible to repeat the treatment

This obvious omittance was corrected and information added: “Makeup removal began with the use of a Q switch nanosecond laser with a wavelength of 1064nm, spot size 4mm, 3.2 J/cm2, 6 Hertz (Laser Xlase Plus from Biotec Italia Poland)” Further sessions’ parameters were also described in detail, also in the Figure 3 all parameters were outlined with accurate description and units.

[pls see new Figure in the attachment]

- There are mistakes in the description of units, such as joules

Corrected.
- The case description does not follow appropriate guidelines such as CARE

More information was added to meet CARE guidelines.  (please see above, and other added sections: “In a follow-up at approximately 6 months post makeup removal, each patient was satisfied with the results and had not experienced any further allergic reactions.”)
- Images/photos of each follow-up should be taken from the same angle

Images were sorted and organized to unify cases presentations.

Apart from these changes some new sections and references and a Table were added /in red/.

Reviewer 2 Report

Comments and Suggestions for Authors

The current manuscript is an interesting case report on severe dermatological complications arising from permanent makeup removal, and it is overall well structured, with representative photographs of every relevant step of the removal procedures. Hence, I only advise on the following alterations before acceptance for publication:

- In author affiliations, affiliation number 1 should be more detailed;

- In the introduction section, more should be said on cosmetic tattooing, or micropigmentation, in  a more detailed manner; a figure representing the procedure, its functions, and potential side effects, as well as removal procedures and their limitations, should also be produced and added, as a schematic representation;

- References are not appearing in order; for example, in the first paragraph of the introduction section, reference 11 is coming after reference 1, whereas reference 2 should come after reference 1;

- In general, more references should be added in order to better support what is being said;

- The sections and subsections should all be adequately numbered;

- The results should be compared to other similar case reports already existing in the scientific literature, with thorough comparative discussion;

- An abbreviation list is missing and should be added.

Author Response

Dear Editor and Reviewers,

Thank you for your interest in our manuscript entitled "Permanent makeup removal severe complication – case report and proposition of the protocol for its management”. We would like to thank Reviewers for their valuable comments and reviews, which helped to improve the manuscript. In this revision we addressed all your comments (in the text changes are marked in red). Along Reviewers’ comments we performed extensive editing of the paper – it was proofread by the native and references were updated and organized. All insertions are marked in red /stylist and grammar changes were not marked as they were too extensive and complex/. We are sorry for not checking this before initial submission. We hope that now, the paper reads well and will meet the requirements of the Reviewers.

Sincerely,

The Authors

Reviewers’ comments:

Reviewer 2

The current manuscript is an interesting case report on severe dermatological complications arising from permanent makeup removal, and it is overall well structured, with representative photographs of every relevant step of the removal procedures. Hence, I only advise on the following alterations before acceptance for publication:

Thank you!

- In author affiliations, affiliation number 1 should be more detailed.

Done.

- In the introduction section, more should be said on cosmetic tattooing, or micropigmentation, in a more detailed manner; a figure representing the procedure, its functions, and potential side effects, as well as removal procedures and their limitations, should also be produced and added, as a schematic representation;

Done. These aspects were expanded in the text and in a new table /added/:

“The main indications for PMU include: an alternative to conventional makeup (lips, eyebrows, eyelines) in order to save time, when its performance is difficult, e.g. due to motoric or visual disabilities, or when a person is allergic to traditional cos-metics. The second indication includes camouflage and/or reconstructive purposes (nipple areola complex, birthmarks, alopecia/vitiligo, and scars) [3].  Complications which may occur after PMU are most often related to the insertion of a microneedle in the skin and include bleeding, itching, and edema, but there are also reports of rare complications sometimes appearing many years after the procedure. These include granulomatous pseudotumor, loss of lashes, margin irregularities, dry eye syndrome, corneal pigmentation, corneal defect/injury, conjunctivitis, granu-lomatous inflammatory reaction, trichiasis, extensive hyperpigmentation of the na-sojugal fold and lower eyelid and even eyelid necrosis, MRI complications, and dissatisfaction with the results [4-12]. The study performed by Tomita et al. (2021) included 1352 respondents who underwent permanent makeup procedures for the eyebrow and eyeline. They reported an overall complication rate of 12.1%. However, in most cases complications were related to the reaction of needle passage: itching of eyebrow (8.2%) and swelling of eye-lined skin (13.2%). The most severe complica-tions were local infections (0.2%). No allergies to the used inorganic pigments were detected [13]. When the result of PMU is not satisfactory

but permanent, there are two main methods of PMU removal used: laser therapy and/or

chemical remover. Their effectiveness is dependent on the pigments used (Table 1).”

Table 1 Methods used for permanent eyebrow makeup removal and/or correction. *

[please see a new Table in the attachment]

- References are not appearing in order; for example, in the first paragraph of the introduction section, reference 11 is coming after reference 1, whereas reference 2 should come after reference 1;

Sure! Sorry about the mess in references – corrected!

- The sections and subsections should all be adequately numbered;

Done!

- In general, more references should be added in order to better support what is being said;

- The results should be compared to other similar case reports already existing in the scientific literature, with thorough comparative discussion;

Such section discussing other similar cases was added in Discussion. Reference list was updated! E.g. a section was added:

Contrary to red ink tattoo, there are not many reports concerning PMU allergic reactions. Such four cases were described by Wenzel et al. [30] in whose patients severe and therapy-resistant skin reactions appeared after lip PMU with the use of Pigment Red. The treatment included topical steroids and tacrolimus, which appeared not to be fully effective. The Authors recommended the regulation and control of the substances used in PMU colorants [30]. Goldman and Wollina [26] described a massive displacement of micropigments following eyelid tattooing which was treated with three consecutive sessions of a Q-switched 1,064 nm neodymium-doped yttrium aluminum garnet (Nd:YAG) laser – the course of treatment was uneventful and effective. The fact that in our cases allergic reaction appeared could have been related to the different type of pigments and different mixture of pigments including not only black ink. We have not found a case of allergic reaction induced by PMU removal with laser. Such reactions, even anaphylactic, were however reported in cases of tattoo removals, red-inked but also with only black [31,32,33]. The presented case points to the possibility of similar reactions in cases of PMU in the periorbital area.”

- An abbreviation list is missing and should be added.

Done.

Reviewer 3 Report

Comments and Suggestions for Authors

Dear Authors,

I write you in regard to your manuscript entitled "Permanent makeup removal severe complication – case report and proposition of the protocol for its management". 

- please, add and distribute references in the introduction.

- please, try to describe the chemicals used and the medications as well, discussing their risks and benefits. possible choices of drugs etc. 

- please, register the approval of the local ethics committee.

Overall, this is an interesting case report.

Author Response

Dear Editor and Reviewers,

Thank you for your interest in our manuscript entitled "Permanent makeup removal severe complication – case report and proposition of the protocol for its management”. We would like to thank Reviewers for their valuable comments and reviews, which helped to improve the manuscript. In this revision we addressed all your comments (in the text changes are marked in red). Along Reviewers’ comments we performed extensive editing of the paper – it was proofread by the native and references were updated and organized. All insertions are marked in red /stylist and grammar changes were not marked as they were too extensive and complex/. We are sorry for not checking this before initial submission. We hope that now, the paper reads well and will meet the requirements of the Reviewers.

Sincerely,

The Authors

Reviewers’ comments:

Reviewer 3

I write you in regard to your manuscript entitled "Permanent makeup removal severe complication – case report and proposition of the protocol for its management". 

- please, add and distribute references in the introduction.

Sure! Sorry about the mess in references – corrected!

- please, try to describe the chemicals used and the medications as well, discussing their risks and benefits. possible choices of drugs etc. 

This was done – remover was characterized in detail and discussed. Also a table was added to outline a possible treatments for PMU removal complications’ treatment:

  Table 1 Methods used for permanent eyebrow makeup removal and/or correction. *

[please see a new Table in the attachment]

- please, register the approval of the local ethics committee.

Included. “The study was approved by the Bioethical Committee of the Medical University of Lodz (RNN/173/22/KE).”

Overall, this is an interesting case report.

Thank you!

Round 2

Reviewer 3 Report

Comments and Suggestions for Authors

Dear Authors,

Thank you for the revised and improved version of your manuscript. Please, consider revising the Figures, if applicable.